# Tryptophan Metabolism in Postmenopausal Women with Functional Constipation

**DOI:** 10.3390/ijms25010273

**Published:** 2023-12-24

**Authors:** Aleksandra Blonska, Marcin Chojnacki, Anna Macieja, Janusz Blasiak, Ireneusz Majsterek, Jan Chojnacki, Tomasz Poplawski

**Affiliations:** 1Department of Clinical Nutrition and Gastroenterological Diagnostics, Medical University of Lodz, 90-647 Lodz, Poland; aleksandra.blonska@umed.lodz.pl (A.B.); marcin.chojnacki@umed.lodz.pl (M.C.); 2Department of Pharmaceutical Microbiology and Biochemistry, Medical University of Lodz, 92-215 Lodz, Poland; anna.macieja@umed.lodz.pl; 3Faculty of Medicine, Collegium Medicum, Mazovian Academy in Plock, 09-402 Plock, Poland; j.blasiak@mazowiecka.edu.pl; 4Department of Clinical Chemistry and Biochemistry, Medical University of Lodz, 92-215 Lodz, Poland; ireneusz.majsterek@umed.lodz.pl; 5Biohazard Prevention Centre, Faculty of Biology and Environmental Protection, University of Lodz, 90-236 Lodz, Poland

**Keywords:** functional constipation, menopause, tryptophan, 5-hydroxyindoleacetic acid, kynurenine, indican

## Abstract

Constipation belongs to conditions commonly reported by postmenopausal women, but the mechanism behind this association is not fully known. The aim of the present study was to determine the relationship between some metabolites of tryptophan (TRP) and the occurrence and severity of abdominal symptoms (Rome IV) in postmenopausal women with functional constipation (FC, n = 40) as compared with age-adjusted postmenopausal women without FC. All women controlled their TRP intake in their daily diet. Urinary levels of TRP and its metabolites, 5-hydroxyindoleacetic acid (5-HIAA), kynurenine (KYN), and 3-indoxyl sulfate (indican, 3-IS), were determined by liquid chromatography/tandem mass spectrometry. Dysbiosis was assessed by a hydrogen–methane breath test. Women with FC consumed less TRP and had a lower urinary level of 5-HIAA, but higher levels of KYN and 3-IS compared with controls. The severity of symptoms showed a negative correlation with the 5-HIAA level, and a positive correlation with the 3-IS level. In conclusion, changes in TRP metabolism may contribute to FC in postmenopausal women, and dysbiosis may underlie this contribution.

## 1. Introduction

Functional constipation (FC) is a common disorder of the gastrointestinal tract, with an average prevalence of about 10% if diagnosed according to Rome IV criteria [1]. This relatively high prevalence results mainly from bad eating habits, lack of physical exercise, low hydration, aging, and several other factors, and most often a combination of them [2]. According to Rome IV Criteria (https://theromefoundation.org/rome-iv/rome-iv-criteria/ (accessed 12 December 2023)), disorders of chronic constipation can be classified into four subtypes: (a) functional constipation; (b) irritable bowel syndrome (IBS) with constipation; (c) opioid-induced constipation; and (d) functional defecation disorders, including inadequate defecatory propulsion and dyssynergic defecation. Diagnostic criteria of FC are straining for more than 1/4 time of defecations, lumpy or hard stools, the sensation of incomplete evacuation, the sensation of anorectal obstruction/blockage, manual maneuvers to facilitate defecation, less than three spontaneous bowel movements per week, and loose stools rarely present without the use of laxatives. The symptoms should not meet Rome IV Criteria for IBS [3]. Abdominal pain is a symptom differentiating FC from IBS, as it occurs less often in FC and has no temporal connection with the act of defecation. There are three types of FC: constipation with normal bowel movements, constipation with slow bowel movements, and rectal emptying disorders. The underlying mechanisms of these forms of constipation are still poorly understood, but it is generally accepted that they are multifactorial in nature. Psychophysiological factors for FC include, but are not limited to, genetic burdens, psychiatric disorders, abnormal bowel motility, poor diet, abnormalities in the intestinal microbiome, neurotransmitter imbalances, and hormonal homeostasis [1,2].

Hormonal homeostasis presumably plays a predominant role in the occurrence of FC in postmenopausal women, when the female body undergoes complex changes associated with hormonal alterations [4]. For this reason, postmenopausal women face numerous psychosomatic complaints related to the gastrointestinal tract, such as eating disorders, abdominal bloating, and chronic constipation, seriously affecting their quality of life [5]. However, FC occurs only in a fraction of postmenopausal women, indicating the involvement of factors other than hormonal imbalances that are not fully known, but indirect evidence points at the serotonin (5-HT) pathway of tryptophan (TRP) metabolism. Estrogen was shown to improve perimenopausal symptoms by increasing the 5-HT pathway and modulating its neurotransmission in the central nervous system [6,7,8,9,10,11].

Consistent evidence suggests that postmenopausal women are more susceptible to changes in TRP metabolism than their counterparts before menopause [12]. Most TRP (over 90%) is metabolized in the digestive tract in the serotonin (1–2%), kynurenine (approximately 95%) and indole (2–3%) pathways, and their initiating enzymes are TRP hydroxylase (TPH-1), indoleamine 2,3-diooxygenase (IDO-1), and bacterial TRPase (TNA), respectively. These enzymes compete for access to ingested TRP [13]. In healthy people, the balance of these pathways is maintained, but it can be disturbed by many factors, including TRP intake and changes in the gut microbiota [14,15]. In these cases, the relationship between TRP intake and the activity of the kynurenine and indole pathways in postmenopausal women has not been determined. Serotonin is mainly produced in the intestine, where it regulates motility and secretion. Both high and low levels of 5-HT have been found in functional intestinal disorders, and patients with constipation often show reduced levels of 5-HT [16,17,18,19]. Therefore, 5-HT homeostasis may play an important role in the pathogenesis of chronic constipation and consequently may be useful in the prevention, diagnosis, and treatment of this disease.

The aim of the present study was to determine the relationship between TRP, along with its metabolites, and the severity of abdominal symptoms in postmenopausal women with functional constipation. In general, studies on TRP metabolism in menopause are scarce, and to our knowledge, this is the first study linking TRP metabolism and constipation in postmenopausal women. Therefore, our work fits a literature gap, but on the other hand, it aims to provide information about possible management of menopausal constipation with changes in the diet.

## 2. Results

Table 1 presents the results of routine laboratory tests in postmenopausal women with and without functional constipation. Both groups differ significantly in the levels of follicular-stimulating hormone (*p* < 0.05). All other characteristics do not differentiate between these two groups. However, dietary TRP intake was significantly lower in postmenopausal women with FC.

Hydrogen concentrations in exhaled air at the start of the hydrogen/methane breath test (time 0) and after 90 min were higher in the patient group than controls (Table 2, *p* < 0.01 and *p* < 0.001, respectively). There was no difference between patients and controls after 180 min of the test. Methane concentration was higher in patients than controls after 90 and 180 min (Table 2, *p* < 0.05), but did not exceed acceptable limits.

Urinary levels of TRP and 5-HIAA were lower for postmenopausal women with FC than their counterparts without FC (Figure 1, *p* < 0.05).

Urinary levels of KYN and 3-IS were higher in the patient group (Figure 2, *p* < 0.001).

The correlation analysis of the intensity of symptoms with female hormones, TRP intake, urinary levels of TRP and its metabolites KYN, 5-HIAA and 3-IS showed a negative correlation for 5-HIAA and a positive correlation for 3-IS (Table 3, Figure 3).

## 3. Discussion

Menopause is associated with a series of unpleasant symptoms that affect postmenopausal women. Identification of these symptoms and the mechanisms behind them may allow us to prevent and ameliorate their consequences. In the present work, we aimed to determine if tryptophan and its metabolism may contribute to mechanisms underlying functional constipation in postmenopausal women.

We did not observe any correlation between the concentration of female sex hormones and the intensity of abdominal symptoms associated with chronic functional constipation in postmenopausal women. The same was true for TRP intake, although it was significantly lower in women suffering from constipation. We did observe differences in urinary levels of TRP and its metabolites, KYN, 5-HIAA, and 3-IS: TRP and 5-HIAA levels decreased, while KYN and 3-IS increased, in postmenopausal women with constipation. This relationship may reflect a decreased activity of serotonin and increased activity of kynurenine and the indole pathways of TRP metabolism. Furthermore, the severity of abdominal symptoms showed a negative correlation with urinary levels of 5-HIAA and a positive correlation with urinary levels of 3-IS. Higher levels of hydrogen and methane in the breath test and 3-IS in the urine of women with FC suggest a bacterial overgrowth, mainly in the colon. Indican is synthesized through colon microbes on the TRP–indole–3-indoxyl sulfate pathway, and it is considered a quantitative biomarker of the intestinal microbiome. Other studies also found changes in the composition of the intestinal microbiome and its metabolites in FC. Short-chain fatty acids (SCFA), secondary bile salts, and methane can affect gut motility and secretion in patients with FC [20,21]. These factors act through the activation of appropriate receptors that contribute in some enteroendocrine and neuronal cells to synthesize bioactive compounds, such as neurotransmitters and peptides.

Short-chain fatty acids, including indoleacetic acid (ILA), indolepropionic acid (IPA), and butyric acid (EAA), are mainly produced by colonic anaerobic bacterial fermentation of dietary compounds. They regulate physiological functions, including colonic blood flow, fluid-end electrolyte reuptake, and modulation of gut motility [22,23]. Increased butyric acid could contribute to constipation through a decrease in some bacteria species, but too high SCFA concentrations could lead to diarrhea [24,25]. Other studies reported that acetic acid and propionic acid increased gut motility or inhibited it [26,27,28,29].

Methane is generated by methanogens (bacteria and archaea) in the colon, but it is also produced by anaerobic fermentation of indigested polysaccharides [30]. It can act on the gut motility via changes in 5-HT concentration, and its production can correlate with a lower number of bowel movements [31,32,33]. However, another study indicated that methane production was not associated with constipation [34] and patients with normal gut transit produced more methane compared to controls [35]. In our study all menopausal women had increased levels of exhaled hydrogen and methane. Therefore, further studies are needed to link constipation-related pathogenic mechanisms of methane with SCFA.

The gut microbiota promotes the biosynthesis of 5-HT in the host by regulating TRP metabolism. 5-HT is involved in smooth muscle contraction or relaxation and regulates its function via various receptors. In animals, gut dysbiosis was shown to upregulate the expression of serotonin transporter (SERT) and decrease 5-HT concentration in the colon mucosa, which weakened the intestinal circular muscle concentration activity and inhibited intestinal motility [36]. Other studies reported that increased 5-HT concentration in the blood or colonic mucosa was related to constipation [37,38]. These differences are likely due to the balance between receptor activation and desensitization in the outcome of gastrointestinal motility. These processes can be affected by the composition of the intestinal microbiota. Many studies showed that the alteration of the gut microbiome was associated with constipation [39,40]. A number of clinical studies indicate that the composition of the intestinal microbiota differs between constipated patients and healthy individuals [24,34,41,42]. The gut microbiota regulates 5-HT production through several mechanisms, including its impact on enterochromaffin cell growth and TRP metabolism [43,44,45]. It is suggested that the gut bacteria may also upregulate the production of KYN and tryptophan indole, thus reducing the substrate to produce 5-HT [46]. Such a mechanism might occur in our patients. It should be noted that most of the results cited above are related to preclinical studies, and findings from animals may be incoherent with those from humans. However, several clinical studies confirm the importance of dysbiosis in the pathogenesis of chronic constipation by regulating SERT in the intestine [36,47]. On the other hand, some species of bacteria have the ability to abundantly produce 5-HT [48]. These complex mechanisms confirm the beneficial effect of fecal microbiota transplantation from healthy donors on patients suffering from functional diarrhea and constipation [49,50].

In medical practice, probiotics are often used to treat functional diseases of the gastrointestinal tract, beneficially altering the gut microbiota and relieving abdominal symptoms, including constipation [51,52,53]. Probiotics may improve intestinal transit time, stool consistency, and frequency [54,55,56]. Their multi-species structure is more effective than that of single species [57]. In the case of intestinal bacterial overgrowth, it is justified to use antibiotics such as rifaximin or metronidazole. Rifaximin can modulate intestinal microbial composition through eubiotic activity [58] and accelerate colonic transit [59]. Dietary fibers and TRP may also relieve constipation by optimizing the gut microbiome [60,61].

All patients and controls underwent the hydrogen/methane breath test to assess bacterial overgrowth. However, we did not analyze the microbiota composition of the subjects. Chronic constipation may occur if the intestinal luminal environment, including microbiota composition, is disturbed, and, in general, this reflects a disturbance in the regulation of the gut–brain–microbiota axis (reviewed in [62,63]). Therefore, the determination of the microbiota composition should be included in further research to determine the specificity of the observed symptoms. The subjects enrolled in our study did not report taking any probiotics, at least for the last 6 months before the study. Therefore, we have not considered probiotic administration as a confounding factor in our analysis. However, many probiotics were reported to play a role in relieving constipation; therefore, the history of taking probiotics should be included in the analysis in further studies as an important factor that can modulate constipation (reviewed in [64]).

We obtained data on some aspects of the TRP metabolism in the population of postmenopausal women, whose part experienced the specific symptom—chronic constipation. Therefore, we provided a general direction for possible further research—TRP metabolism in menopause. As TRP metabolism may be, at least in part, regulated by changes in dietary tryptophan content, such an easy-to-implement dietary intervention might lead to the relaxation of aggravating symptoms of menopause. There are several limitations to our study. The cohort we studied was relatively small—40 controls and 40 patients. We concluded that dysbiosis is a possible mechanism behind constipation, but we did not perform an analysis of the composition of the intestinal microbiota. Another limitation may be the non-homogenous age of patients, as menopausal symptoms may be age-dependent [65]. The same concerns obesity, because vasomotor symptoms frequently experienced by women have been associated with obesity, and it is suggested that weight management efforts may reduce the severity of menopausal symptoms [66]. There were no differences between patients and controls in our study, and there were no obese (BMI of 30 or greater) subjects either.

Dietary therapy is still the basis of FC treatment. The diet should contain an appropriate amount of fiber but also other ingredients, including TRP. In constipation patients, it is recommended to determine TRP metabolism because the metabolites of all its pathways may be the cause of both digestive and mental disorders [67,68]. Further research is necessary to determine the strains of bacteria that metabolize TRP. This will facilitate quantification for the probiotic treatment of functional gastrointestinal diseases. Our study also suggests a link between menopause, constipation, and tryptophan metabolism, but its details are largely unknown and should be studied in further research.

## 4. Methods and Materials

### 4.1. Patients

This study included 80 postmenopausal women, aged 57–66 years, recruited in 2018–2022. During the recruiting procedure, all individuals were tested for SARS-CoV-2 infection, and only those with a negative result were enrolled. Two groups were distinguished of 40 women each: controls—women without any digestive complaints; and patients—women with functional constipation. Inclusion criteria included the following: the last menstruation at least 2 years earlier; a diagnosis of functional constipation consistent with Rome IV Criteria (https://theromefoundation.org/rome-iv/rome-iv-criteria/ (accessed 12 December 2023)); other diseases of the gastrointestinal tract; and the use of antispasmodic, hormonal, and psychotropic drugs. Both patients and controls did not report taking any antibiotic/probiotic in the 6 months before the study. All individuals underwent imaging and endoscopic tests. Subjects with inflammatory bowel diseases, celiac disease, allergy, and food intolerance, parasitic infestations, thyroid diseases, diabetes and other metabolic diseases, mental disorders, and taking hormones and psychotropic drugs were excluded from this study.

Prior to the study, informed consent was obtained from all subjects. The study was conducted according to the guidelines of the Declaration of Helsinki and the Guidelines for Good Clinical Practice and approved by the Bioethics Committee of the Medical University of Lodz (RNN/176/18/KE).

### 4.2. Diagnostic Procedures

The severity of abdominal symptoms included in the Rome IV Criteria and mentioned earlier was assessed and scored between 1 and 7 points. The state of the gut microbiome was assessed by the hydrogen/methane breath test using gas chromatography on a GastroCH4ECK (Bedfont Scientific LTD, Harrietsham, Maidstone, UK). Hydrogen and methane contents in the breath were measured in 15 min intervals within 3 h, after consuming 10 g of lactulose dissolved in 200 mL of water. Values obtained at 0, 90, and 180 min were used for statistical analysis.

### 4.3. Laboratory Tests

The following routine laboratory tests were performed in all subjects: blood cell count, protein quantification, glucose, glycated hemoglobin, profile of lipids, bilirubin, iron, urea, creatinine, glomerular filtration rate, thyroid stimulating hormone, free thyroxine, free triiodothyronine, amylase, lipase, alanine and asparagine aminotransferase, alkaline phosphatase, gamma-glutamyltranspeptidase, deaminated gliadin peptide, antibodies against tissue transglutaminase, and fecal calprotectin. Moreover, 17-β-estradiol (Ortho antibodies—Clinical Diagnostics kit) and follicular-stimulating hormone (FSH—Vitros Product antibodies) were determined.

Urine samples for TRP and its metabolites were collected in the morning into a special container with a 0.1% hydrochloric acid solution as a stabilizer. The concentration of TRP and its following metabolites: 5-hydroxyindoleacetic acid (5-HIAA), kynurenine (KYN), and 3-indoxyl sulfate (3-IS) were determined using liquid chromatography with tandem mass spectrometry (LC-MS/MS) according to the manufacturer’s instructions (Ganzimmun Diagnostics AG, Mainz, Germany). The levels of these metabolites were expressed in mg per gram of creatinine (mg/gCr). The semetabolites of TRP were considered exponents of the activity of serotonin, kynurenine, and indole pathways of TRP metabolism. All laboratory materials and the results of the breath test were collected on the same day.

### 4.4. Nutritional Recommendations

All individuals were recommended to record the type and quantity of products consumed every day for 14 days prior to investigation in the nutritional diary. The average daily TRP intake was calculated using the nutritional calculator with the Kcalmar.pro-Premium application (Hermex, Lublin, Poland). On the day before the evaluation, everyone received a diet with the TRP content calculated individually in advance.

### 4.5. Data Analysis

The Shapiro–Wilk W test was used to assess the normality of the distribution of variables. General biochemical parameters of both controls and patients were normally distributed, and the Student’s *t*-test was used to compare differences between controls and patients. Lack of normality in the distribution of TRP, KYN, 5-HIAA, and 3-IS resulted in the use of nonparametric tests to assess differences between the two groups (the Mann–Whitney U test) and correlation analysis (the Spearman rank test). All statistical analyses were performed with STATISTICA 13.3 software (TIBCO Software Inc., Palo Alto, CA, USA).

## Figures and Tables

**Figure 1 ijms-25-00273-f001:**
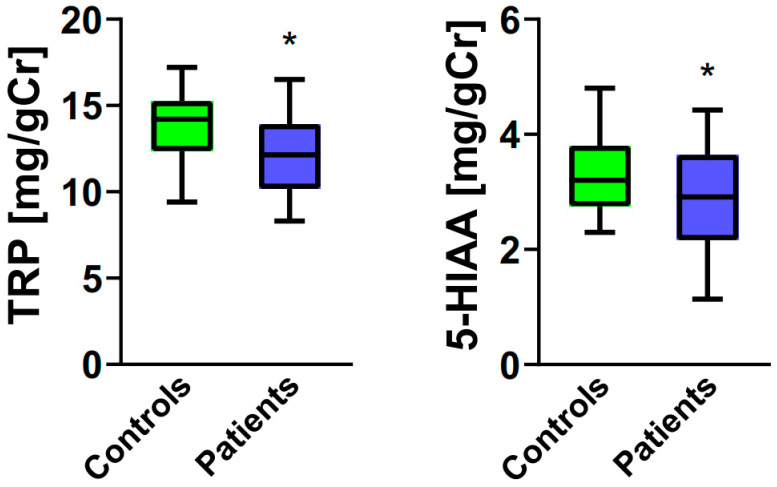
Urinary levels of tryptophan (TRP, **left panel**) and 5-hydroxyindoleacetic acid (5-HIAA, **right panel**) expressed in milligrams per gram of creatinine (mg/gCR) in postmenopausal women (controls, n = 40) and postmenopausal women with functional constipation (patients, n = 40). Median with boxes representing I and III quartiles, and error bars represent minimal and maximal values. Differences between patients and controls were evaluated by the Mann–Whitney U test; *—*p* < 0.05.

**Figure 2 ijms-25-00273-f002:**
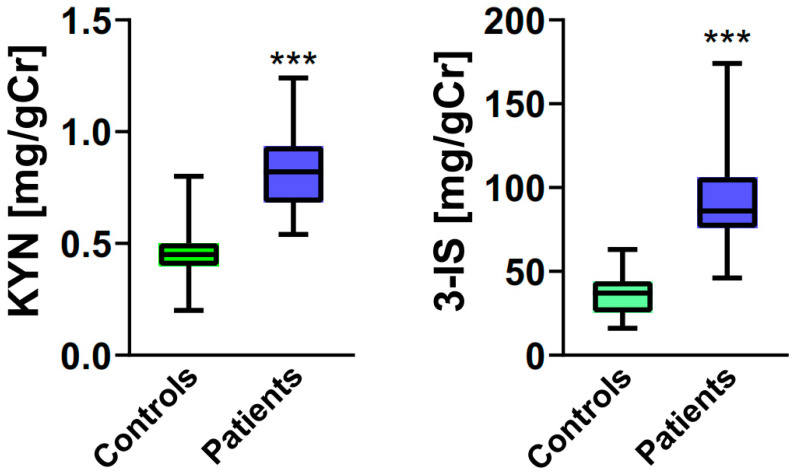
Urinary levels of kynurenine (KYN, **left panel**) and 3-indoxyl sulfate (3-IS, **right panel**) expressed in milligrams per gram of creatinine (mg/gCR) in postmenopausal women (controls, n = 40) and postmenopausal women with functional constipation (patients, n = 40). Median with boxes representing I and III quartiles, and error bars represent minimal and maximal values. Differences between patients and controls for both compounds were evaluated by the Mann–Whitney U test; ***—*p* < 0.001.

**Figure 3 ijms-25-00273-f003:**
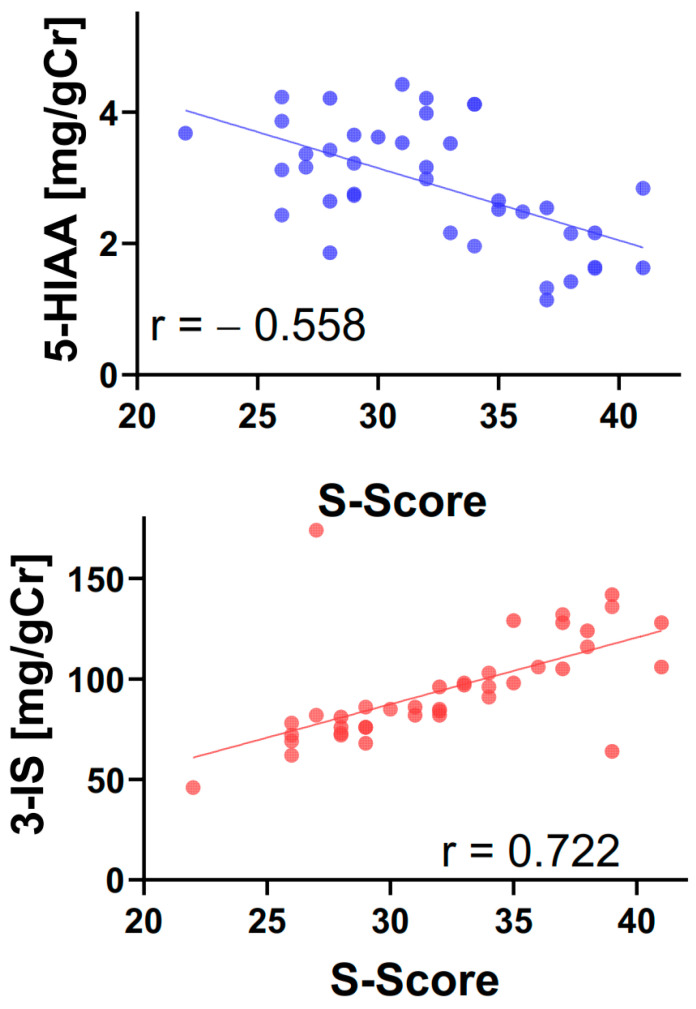
Correlation between the severity of symptoms (S-score) and the urinary levels of 5-hydroxyindoleacetic acid (5-HIAA) and 3-indoxyl sulfate (3-IS) in postmenopausal women with functional constipations (n = 40). The correlations were analyzed using the Spearman rank test with the rho rank correlation coefficient (r).

**Table 1 ijms-25-00273-t001:** Characteristics of control postmenopausal women (controls, n = 40) and postmenopausal women with functional constipation (patients, n = 40) enrolled in this study.

Feature ^a^	Controls	Patients	*p*
Age (years)	58.6 ± 7.6	57.9 ± 7.4	ns
BMI (kg/m^2^)	25.4 ± 2.3	24.8 ± 1.7	ns
GFR (mL/min)	98.6 ± 5.9	97.1 ± 4.5	ns
ALT (µ/L)	15.6 ± 3.5	14.2 ± 3.2	ns
AST (µ/L)	13.9 ± 2.1	14.3 ± 2.5	ns
CRP (mg/L)	2.3 ± 1.8	3.1 ± 2.2	ns
FC (µg/g)	26.6 ± 15.7	30.9 ± 12.8	ns
ES (pg/mL)	19.6 ± 4.7	18.8 ± 4.1	ns
FSH (UI/mL)	108.9 ± 22.4	119.8 ± 24.6 *	<0.05
TRP intake (mg daily)	1163 ± 125	1065 ± 90	<0.01

^a^ mean ± SD (standard deviation); BMI—body mass index; GFR—glomerular filtration rate; ALT—alanine aminotransferase, AST—aspartate aminotransferase, CRP—C-reactive protein; FC—fecal calprotectin; ES—17-β-estradiol; FSH—follicular-stimulating hormone; differences between groups were assessed by Student’s *t*-test; *–*p* < 0.05; ns—non-significant (*p* > 0.05).

**Table 2 ijms-25-00273-t002:** Concentrations of hydrogen and methane in exhaled air of postmenopausal women (controls, n = 40) and postmenopausal women with functional constipation (patients, n = 40) determined at 0, 90, and 180 min with the hydrogen/methane test.

Chemical (Time, min)	Controls (ppm)	Patients (ppm)	*p* ^a^
Hydrogen (0)	6.75 ± 2.14	14.7 ± 5.72 **	<0.01
Hydrogen (90)	23.1 ± 5.72	29.8 ± 9.63 ***	<0.001
Hydrogen (180)	93.4 ± 19.8	99.1 ± 15.8	ns
Methane (0)	4.7 ± 1.6	4.7 ± 1.3	ns
Methane (90)	4.6 ± 1.1	5.2 ± 1.2 *	<0.05
Methane (180)	12.1 ± 4.1	15.4 ± 5.6 **	<0.05

^a^ differences between groups were assessed by the Student’s *t*-test; *–*p* < 0.05; **–*p* < 0.05; ***–*p* < 0.001; ns—non-significant (*p* > 0.05).

**Table 3 ijms-25-00273-t003:** Correlation between the severity of symptoms (S-score) and serum estrogen (E-2) and follicular-stimulating hormone (FSH) levels, tryptophan (TRP) intake, and urinary levels of TRP (TRP), 5-hydroxyindoleacetic acid (5-HIAA), kynurenine (KYN), and 3-indoxyl sulfate (3-IS) in postmenopausal women with functional constipations (n = 40). The correlations were analyzed using the Spearman rank test with the rho rank correlation coefficient.

Pairs of Variables	rho-Spearman	*p*
S-score and E-2	0.0435	ns
S-score and FSH	−0.2897	ns
S-score and TRP intake	0.0517	ns
S-score and TRP	−0.0233	ns
**S-score and 5-HIAA**	**−** **0.5578**	<0.001
S-score and KYN	−0.1552	ns
**S-score and 3-IS**	**0.7215**	<0.001

## Data Availability

Data from this study are ready to be shared upon reasonable request.

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
