# Peer review of "Tryptophan Metabolism in Postmenopausal Women with Functional Constipation"

_ijms, 2023, doi:10.3390/ijms25010273_

Round 1

Reviewer 1 Report

Comments and Suggestions for Authors

1.

The period of study is during COVID pandemic-- there should be COVID parameters to be taken careoff -- need to mention in result and discussion. As these factirs have an effect

2.

The probiotics effect GI tract but not incorporated in experiments-- advised to incorporate probiotics taken history in the material method of the participants. Effect of Microbiota also need to be discussed

Hossain, M. I., Islam, R., Mimi, S. I., Jewel, Z. A., & Haider, U. A. (2020). Gut microbiota: Succinct overview of impacts on human physique and current research status with future aspects. International Journal of Advancement in Life Sciences Research3(2), 1-10.

3. 

The reference numbering not right-- Check  32, 47

4.

Following references need to cite

Chu-wen Ling, Zelei Miao, Mian-li Xiao, Hongwei Zhou, Zengliang Jiang, Yuanqing Fu, Feng Xiong, Luo-shi-yuan Zuo, Yu-ping Liu, Yan-yan Wu, Li-peng Jing, Hong-Li Dong, Geng-dong Chen, Ding Ding, Cheng Wang, Fang-fang Zeng, Hui-lian Zhu, Yan He, Ju-Sheng Zheng, Yu-ming Chen, The Association of Gut Microbiota With Osteoporosis Is Mediated by Amino Acid Metabolism: Multiomics in a Large Cohort, The Journal of Clinical Endocrinology & Metabolism, Volume 106, Issue 10, October 2021, Pages e3852–e3864, https://doi.org/10.1210/clinem/dgab492

Chojnacki, C., Konrad, P., Mędrek-Socha, M., Kaczka, A., Chojnacki, M., & Błońska, A. (2022). Altered tryptophan metabolism in patients with recurrent functional abdominal pain. Polski Merkuriusz Lekarski50(295).

Author Response

Comment: 1. The period of study is during COVID pandemic-- there should be COVID parameters to be taken careoff -- need to mention in result and discussion. As these factirs have an effect

Answer: We have added the following sentence to the Material and Methods section:

“During the recruiting procedure, all individuals were tested for SARS-CoV-2 infection and only persons with a negative result were included in this study.”

Comment: The probiotics effect GI tract but not incorporated in experiments-- advised to incorporate probiotics taken history in the material method of the participants. Effect of Microbiota also need to be discussed Hossain, M. I., Islam, R., Mimi, S. I., Jewel, Z. A., & Haider, U. A. (2020). Gut microbiota: Succinct overview of impacts on human physique and current research status with future aspects. International Journal of Advancement in Life Sciences Research, 3(2), 1-10.

Answer: We have added the following sentence to Methods and Material:

“Both patients and controls did not report taking any antibiotic/probiotic in 6 months before the study.”

We have added the following fragments to the Discussion section:

“All patients and controls underwent the hydrogen-methane breath test to assess a bacterial overgrowth. However, we did not analyze the microbiota composition of the subjects. Chronic constipation may occur if the intestinal luminal environment, including microbiota composition, is disturbed, and in general this reflects the disturbance in the regulation of the gut-brain-microbiota axis (reviewed in [63]). Therefore, determination of the microbiota composition should be included in the further research to determine specificity of the observed symptoms.”

with new reference:

Iancu, M.A.; Profir, M.; Roşu, O.A.; Ionescu, R.F.; Cretoiu, S.M.; Gaspar, B.S. Revisiting the Intestinal Microbiome and Its Role in Diarrhea and Constipation. Microorganisms 2023, 11, doi:10.3390/microorganisms11092177.

and

“The subjects enrolled in our study did not report taking any probiotic at least for the last 6 months before the study. Therefore, we have not considered probiotic administration as a confounding factor in our analysis. However, many probiotics were reported to play a role in reliving constipation, therefore, the history of taking probiotics should be included in analysis in further studies as an important factor that can modulate constipation (reviewed in [64]).”

with new reference

Zhang, T.; Liu, W.; Lu, H.; Cheng, T.; Wang, L.; Wang, G.; Zhang, H.; Chen, W. Lactic acid bacteria in relieving constipation: mechanism, clinical application, challenge, and opportunity. Critical reviews in food science and nutrition 2023, 1-24, doi:10.1080/10408398.2023.2278155.

Comment: The reference numbering not right-- Check  32, 47

Answer: We have reordered the reference numbering in the text and the reference list.

Comment: 4. Following references need to cite Chu-wen Ling, Zelei Miao, Mian-li Xiao, Hongwei Zhou, Zengliang Jiang, Yuanqing Fu, Feng Xiong, Luo-shi-yuan Zuo, Yu-ping Liu, Yan-yan Wu, Li-peng Jing, Hong-Li Dong, Geng-dong Chen, Ding Ding, Cheng Wang, Fang-fang Zeng, Hui-lian Zhu, Yan He, Ju-Sheng Zheng, Yu-ming Chen, The Association of Gut Microbiota With Osteoporosis Is Mediated by Amino Acid Metabolism: Multiomics in a Large Cohort, The Journal of Clinical Endocrinology & Metabolism, Volume 106, Issue 10, October 2021, Pages e3852–e3864, https://doi.org/10.1210/clinem/dgab492Chojnacki, C., Konrad, P., Mędrek-Socha, M., Kaczka, A., Chojnacki, M., & Błońska, A. (2022). Altered tryptophan metabolism in patients with recurrent functional abdominal pain. Polski Merkuriusz Lekarski, 50(295).

Answer: We are sorry, but we do not see any context to cite these references in our manuscript.

Reviewer 2 Report

Comments and Suggestions for Authors

The topic of this research article is quite novel; however, several points need extensive revision before reconsidering it.

- In the abstract, the 2nd sentence needs revision.

- In the abstract, lines 20-23: This sentence should be split into two sentences in order to be more understood to the readers.

- In the abstract, lines 24-25 "...., while a higher .... group". In this sentence a verb is missing.

- Again in the next sentence, there is a similar problem with its last part.

- In the introduction, lines 41-44: This sentence needs english language editing. In its current form, it is not easily understood.

- In the introduction, the literature gap that the present study aims to cover should be emphasized.

- In materials and methods sectin, line 84: A reference should be added concerning the Rome IV criteria.

- In th section 2.5 Data analysis: More details about the statistical tests used should be added (e.g. student t-test? chi-square-test?, ANOVA?).

- p-values should be included in a separate column concerning all tables.

- In figure 1 and figure 2 several data are missing, e.g. p-values, median values, SD or SE, etc.

- Again in figure 3 statistical data should be included, e.g. p-values, r2, etc. These data did not also reported into the text.

- The discussion section need english language editing.

- At the end of the discussion section, the authors should emphasize the strengths of their study and they also reported the limitations of their study.

- The conclusion section is too small. The authors should include more statement about their study, reporting also what studies could be perform in the future based on the results of their study.

- The reference 47 is missing from the references list. Please check the numberin both in the reference list and the text.

- Some more updated references should be added published in the last 2-3 years.

Comments on the Quality of English Language

Extensive editing of English language is strongly required

Author Response

Comment: The topic of this research article is quite novel; however, several points need extensive revision before reconsidering it.

Answer: Thank you.

Comment: - In the abstract, the 2nd sentence needs revision.

Answer: We have combined two first sentences of Abstract:

“Postmenopausal women often have various psychosomatic disorders, including constipation. The causes of which are not fully established.”

into the sentence:

“Constipation belongs to conditions commonly reported by postmenopausal women, but mechanism behind this association is not fully known.”

Comments: - In the abstract, lines 20-23: This sentence should be split into two sentences in order to be more understood to the readers. - In the abstract, lines 24-25 "...., while a higher .... group". In this sentence a verb is missing. - Again in the next sentence, there is a similar problem with its last part.- In the introduction, lines 41-44: This sentence needs english language editing. In its current form, it is not easily understood.

Comment: We have rewritten the whole manuscript, corrected errors and improved its style and we hope that it is much easier to follow in its present form.

Comment: - In the introduction, the literature gap that the present study aims to cover should be emphasized.

Answer: We have added the following fragment in the end of Introduction:

“In general, studies on TRP in menopause are scarce and to our knowledge, this is the first study linking tryptophan metabolism and constipation in postmenopausal women. Therefore, our work fits a literature gap, but on the other side, aims to provide information about possible managing of menopausal constipation with changes in the diet.”

Comment:- In materials and methods sectin, line 84: A reference should be added concerning the Rome IV criteria.

Answer: We have added the following website address:

https://theromefoundation.org/rome-iv/rome-iv-criteria/ (accessed December 12, 2023).

Comment: - In th section 2.5 Data analysis: More details about the statistical tests used should be added (e.g. student t-test? chi-square-test?, ANOVA?).

Answer: Now the section 4.5. Data Analysis is:

“The Shapiro–Wilk W test was used to assess the normality of the distribution of variables. General biochemical parameters of both controls and patients were normally distributed and Student t-test was used to compare differences between controls and patients. Lack of normality in the distribution of TRP, KYN, 5-HIAA and 3-IS resulted in the use of nonparametric tests to assess differences between the two groups (the U Mann-Whitney’s test) and correlation analysis (the Spearman rank test). All statistical analyses were performed with STATISTICA 13.3 software (TIBCO Software Inc., Palo Alto, CA, USA).”

Comment: - p-values should be included in a separate column concerning all tables.

Answer: We have followed that recommendation.

Comments: - In figure 1 and figure 2 several data are missing, e.g. p-values, median values, SD or SE, etc.- Again in figure 3 statistical data should be included, e.g. p-values, r2, etc. These data did not also reported into the text.

Answer: We have provided new figures, with new legends.

Comment- The discussion section need english language editing.

Answer: We have done our best to correct all errors and mistakes in this manuscript and improve its style.

Comment:- At the end of the discussion section, the authors should emphasize the strengths of their study and they also reported the limitations of their study.

Answer: We have added the following fragments to Discussion:

We have added the following sentence to the Material and Methods section:

“Both patients and controls did not report taking any antibiotic/probiotic in 6 months before the study.”

We have added the following fragments to the Discussion section:

“All patients and controls underwent the hydrogen-methane breath test to assess a bacterial overgrowth. However, we did not analyze the microbiota composition of the subjects. Chronic constipation may occur if the intestinal luminal environment, including the microbiota, is disturbed, and in general this reflects the disturbance in the regulation of the gut-brain-microbiota axis (reviewed in [63]). Therefore, determination of the microbiota composition should be included in the further research to determine specificity of the observed syndromes.”

with new reference:

Iancu, M.A.; Profir, M.; Roşu, O.A.; Ionescu, R.F.; Cretoiu, S.M.; Gaspar, B.S. Revisiting the Intestinal Microbiome and Its Role in Diarrhea and Constipation. Microorganisms 2023, 11, doi:10.3390/microorganisms11092177.

and

“The subjects enrolled in our study did not report taking any probiotic at least for the last 6 months before the study. Therefore, we have not considered probiotic administration as a confounding factor in our analysis. However, many probiotics were reported to play a role in reliving constipation, so taking probiotics should be included in further studies as an important factor that can modulate constipation (reviewed in [64])

with new reference

Zhang, T.; Liu, W.; Lu, H.; Cheng, T.; Wang, L.; Wang, G.; Zhang, H.; Chen, W. Lactic acid bacteria in relieving constipation: mechanism, clinical application, challenge, and opportunity. Critical reviews in food science and nutrition 2023, 1-24, doi:10.1080/10408398.2023.2278155.

and

“We obtained data on elements of TRP metabolism in the population of postmenopausal women, which part experienced the specific symptom – chronic constipation. Therefore, we provided general direction of possible further research – TRP metabolism in menopause. As TRP metabolism may be, at least in part, regulated by changes in dietary tryptophan content, such easy to execute dietary intervention might lead to relaxation of aggravating symptoms of menopause. There are several limitations of our study. The cohort we studied was relatively small – 40 controls and 40 patients. We concluded on dysbiosis as a possible mechanism behind constipation, but we did not performed analysis of the composition of the intestinal microbiota. Another limitation may be non-homogenous age of patients as menopausal symptoms may be age-dependent [65]. The same concerns obesity, because vasomotor symptoms are frequently experienced by women have been associated with obesity and it is suggested that weight management efforts may reduce the severity of menopausal symptoms [66]. There were no differences between patients and controls in our study and there were no obese (BMI of 30 or greater) subjects either.”  

with new references

El Khoudary, S.R.; Greendale, G.; Crawford, S.L.; Avis, N.E.; Brooks, M.M.; Thurston, R.C.; Karvonen-Gutierrez, C.; Waetjen, L.E.; Matthews, K. The menopause transition and women's health at midlife: a progress report from the Study of Women's Health Across the Nation (SWAN). Menopause 2019, 26, 1213-1227, doi:10.1097/gme.0000000000001424.

Cao, V.; Clark, A.; Aggarwal, B. Obesity and Severity of Menopausal Symptoms: a Contemporary Review. Curr Diab Rep 2023, doi:10.1007/s11892-023-01528-w.

Comment: - The conclusion section is too small. The authors should include more statement about their study, reporting also what studies could be perform in the future based on the results of their study.

Answer: The conclusion section is not an obligatory element of an IJMS paper, and we have removed it in the revision. Instead, we have included all these elements, mentioned above in Discussion.

Comment: - The reference 47 is missing from the references list. Please check the numberin both in the reference list and the text.

Answer: We have reordered the reference numbering in the text and the reference list.

Comment: - Some more updated references should be added published in the last 2-3 years.

Answer: We have followed that recommendation.

Comment: Comments on the Quality of English Language. Extensive editing of English language is strongly required

Answer: We are sorry for careless language in the original version of this manuscript. We have done our best to correct all errors and mistakes in this manuscript and improve its style.

Round 2

Reviewer 1 Report

Comments and Suggestions for Authors

The article is revised.

2023 reference 63, 64 is added but also advised to add the following

Hossain, M. I., Islam, R., Mimi, S. I., Jewel, Z. A., & Haider, U. A. (2020). Gut microbiota: Succinct overview of impacts on human physique and current research status with future aspects. International Journal of Advancement in Life Sciences Research, 3(2), 1-10. https://doi.org/10.31632/ijalsr.20.v03i02.001

Author Response

Comment: The article is revised.

2023 reference 63, 64 is added but also advised to add the following

Hossain, M. I., Islam, R., Mimi, S. I., Jewel, Z. A., & Haider, U. A. (2020). Gut microbiota: Succinct overview of impacts on human physique and current research status with future aspects. International Journal of Advancement in Life Sciences Research, 3(2), 1-10. https://doi.org/10.31632/ijalsr.20.v03i02.001

Answer: We have added this reference.

Reviewer 2 Report

Comments and Suggestions for Authors

The authors have significantly revised and improved their manuscript. Minor English language editing is recommended.

Comments on the Quality of English Language

Minor English language editing is recommended.

Author Response

Comment: The authors have significantly revised and improved their manuscript. Minor English language editing is recommended.

Answer: We have done our best to improve English in the revised manuscript.